# BEYOND NEXT-TOKEN PREDICTION: DIFFUSION VS. AUTOREGRESSIVE REASONING IN LLMS

## ABSTRACT

We revisit LLM reasoning through two competing decoding paradigms: autoregressive large language models (AR LLMs) with next-token prediction, and diffusion-based large language models (DLLMs) with iterative denoising; yet the community lacks compute-controlled, apples-to-apples comparisons. We recast reasoning as trajectory formation, contrasting sequential commitment in AR LLMs with iterative refinement in DLLMs, and run a matched-scale study across mathematics, logic, natural-language inference, and commonsense QA, with robustness and efficiency analyses. Empirically, DLLMs outperform AR LLMs on most reasoning benchmarks, especially those requiring global constraints and long-range coherence, whereas AR LLMs remain competitive on shorter, commonsense-oriented tasks. Mechanistic analyses show DLLMs gradually correct early errors and enforce sequence-wide consistency; robustness experiments reveal graceful degradation under prompt noise and distribution shift. We quantify accuracy–efficiency trade-offs: DLLMs increase FLOPs and wall-clock latency; compute-matched comparisons preserve their advantage, indicating benefits arise from the generative mechanism rather than added budget. Ablation studies further reveal the influence of diffusion-specific design factors and demonstrate how these parameters affect reasoning performance. While DLLMs incur higher inference cost, our results delineate regimes where diffusion decoding is advantageous and provide practical guidance for model configuration under deployment constraints.

## 1 INTRODUCTION

Large language models (LLMs) have shown strong reasoning across a wide range of tasks. Mainstream systems (*e.g.*, GPT-4 (Achiam et al., 2023), Llama (Touvron et al., 2023), Mistral (Jiang et al., 2023), and DeepSeek (Dai et al., 2024)) use *next-token prediction* with autoregressive, left-to-right decoding. Formally, an AR LLM factorizes the sequence probability into conditional next-token terms and is trained with token-level cross-entropy. This design scales well, aligns with hardware and streaming constraints, and yields predictable inference latency, especially at scale. Yet its virtues also crystallize failures: early choices are irreversible, exposure bias compounds along the chain, and local normalization can induce "myopia," *i.e.*, steps that are locally plausible but globally inconsistent. Recent *de facto* Chain-of-Thought supervision helps by injecting intermediate reasoning, but the underlying process still predicts one token at a time (see Fig. 1(a)).

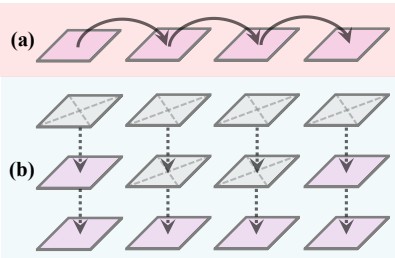

Figure 1: (a) **Autoregressive generation**: outputs are produced sequentially → one token ▱ at a time. (b) **Diffusion-based generation**: text is gradually refined from noise ▱ through iterative ⋯▹ denoising steps.

Inspired by the success of their counterparts in other domains, DLLMs (Li et al., 2022; Karimi Mahabadi et al., 2024) are emerging as a compelling alternative for reasoning. Instead of generating text sequentially like autoregressive models, a diffusion-style model starts from noise and iteratively denoises a representation toward a high-probability sequence (see Fig. 1(b)). If AR decoding is like "speaking," DLLM is like "editing," allowing the model to revise its reasoning trajectory. This provides two key theoretical advantages: reversibility, where late steps can correct early mistakes, and

coarse-to-fine planning, which allows the model to first establish the high-level structure before filling in details. Recent scale-up DLLMs (Ye et al., 2025b; Nie et al., 2025) have shown this paradigm can match or even surpass AR baselines on general tasks. However, operational challenges remain, including a nuanced trade-off between quality and compute governed by discretization strategy, guidance strength, and so on, which directly impacts inference latency.

Despite significant momentum, the community lacks a timely and comprehensive analysis that compares these two paradigms specifically as mechanisms for reasoning, rather than just as alternative decoders. Existing reports (Ye et al., 2024; Deschenaux & Gulcehre, 2024; Feng et al., 2025; Gulrajani & Hashimoto, 2023) vary widely in their benchmarking, metrics, and decoding schedules, making direct, apples-to-apples comparisons difficult. We therefore explicitly frame LLM reasoning as trajectory formation. In this view, AR LLMs construct a discrete trajectory with committed steps, while DLLMs iteratively refine a sequence of partially masked states via re-masking and resampling until convergence. This perspective enables a feasible, systematic comparison across accuracy, robustness, and efficiency. Motivated by this framework, we pose three research questions:

> **RQ1:** *What* is the performance gap between autoregressive LLMs and DLLMs in reasoning across representative tasks under matched model scales?
> **RQ2:** *When* do DLLMs outperform autoregressive LLMs, and *why*?
> **RQ3:** *What* settings (*e.g.*, guidance strength, schedule length, discretization and related parameters) enable DLLMs to facilitate their performance across different winning scenarios?

These research questions **comprehensively review DLLM**, as a new alternative to AR LLM, highlighting its methodological distinctions, practical advantages, and potential to reshape future directions in LLM reasoning. Our contributions and anticipated benefits include: ① **A first-of-its-kind same-scale evaluation.** We perform a systematic, same-scale comparison of DLLMs and AR LLMs of similar size (7∼8B) across 20 diverse reasoning tasks. Our findings reveal that DLLMs yield higher accuracy on tasks requiring global constraints, while LLMs remain competitive on simpler commonsense QA. This work provides an empirical guide on when DLLMs offer gains and when AR LLMs is sufficient, thus supporting informed model choice under deployment needs. ② **An empirical analysis of DLLM's advantages.** Through fine-grained case studies, we provide an analysis of why DLLMs outperform LLMs on certain benchmarks. Our findings show that DLLMs' iterative refinement effectively corrects errors, enforces sequence-wide consistency, and avoids local traps that mislead LLMs. Furthermore, we demonstrate that DLLMs are more robust to prompt noise and distribution shift, revealing that their parallel refinement paradigm enables a level of robustness and global coherence that is inherently difficult to achieve with AR decoding. ③ **Systematic insights into efficiency and design.** We present a detailed investigation into the key design parameters of DLLMs, including remasking strategy, guidance strength, and schedule length, along with their computational and latency trade-offs. The results show that DLLM's performance is maintained in compute-matched settings. This provides a practical guide with principles for balancing accuracy and latency and identifies where diminishing returns begin to appear in the design space.

In short, this paper presents a head-to-head comparison of autoregressive and diffusion-based paradigms for reasoning. The rest of the paper is organized as follows. In §2, we review relevant literature. We then provide some preliminary knowledge regarding the two LLM reasoning paradigms in §3. Our major empirical analysis is presented in §4∼5, and we conclude with a discussion and an outlook on future work in §7.

## 2 RELATED WORK

### 2.1 AUTOREGRESSIVE REASONING

Autoregressive large language models (AR LLMs) decode left-to-right, predicting one token at a time conditioned on the current prefix (Vaswani et al., 2017; Brown et al., 2020; Touvron et al., 2023; Jiang et al., 2023; Du et al., 2022). The procedure is operationally simple and hardware-efficient: computation factorizes into a chain of conditional predictions, latency grows approximately linearly with sequence length, and mature serving stacks deliver high throughput at scale (Narayanan et al., 2021; Rajbhandari et al., 2020; Korthikanti et al., 2022). As a result, AR decoding is the efficiency baseline for long-form inference (Achiam et al., 2023).

The same sequential commitment creates characteristic liabilities (Finlayson et al., 2024). Because decoding is unidirectional and effectively irreversible, early missteps propagate forward and are hard to repair; small local slips can induce long-range inconsistencies and brittle chains of reasoning (Schmidt, 2019; Ranzato et al., 2016; Bengio et al., 2015). Contemporary practice therefore leans on methods such as *test-time scaling*: spending additional inference compute to sample multiple rollouts, elicit intermediate steps, rerank candidates, or verify solutions with external checks (Wang et al., 2025b; Yu et al., 2025; Kumar et al., 2025; Hao et al., 2025; Chen et al., 2024; Joshi et al., 2025; Hao et al., 2025). These procedures improve robustness but remain auxiliary to the core mechanism—they patch rather than remove the fragility of strictly sequential next-token prediction (Zhang et al., 2023; 2024c). Conceptually, AR reasoning resembles human System 1, which aims for fast and intuitive responses when the initial trajectory is sound, yet vulnerable to systematic error when early cues mislead (Chang et al., 2024; Liu & Thoma, 2024; Wu et al., 2024b).

## 2.2 DIFFUSION-BASED REASONING

Diffusion-based language models (DLLMs) introduce a distinct generative paradigm centered on iterative denoising at inference (Sahoo et al., 2024; Chung et al., 2025; Do et al., 2025; Zheng et al., 2023; Austin et al., 2021; He et al., 2023). In contrast to one-pass AR decoding, DLLMs update multiple positions in parallel and repeatedly reconsider the global structure of their output. This provides a principled mechanism for self-correction, allowing the model to overwrite earlier commitments when accumulating evidence contradicts them (Xu et al., 2025; Cardei et al., 2025; Wu et al., 2025; Sahoo et al., 2025). The annealed nature of the denoising process naturally facilitates a coarse-to-fine planning strategy, where early steps focus on high-level semantic structure and later steps fill in details (Zhou et al., 2024a; Karimi Mahabadi et al., 2024; Huang & Tang, 2025). This hierarchical refinement is particularly advantageous for tasks requiring long-range planning, where a global understanding is critical to avoiding local inconsistencies (Lovelace et al., 2023; Xiong et al., 2024). The ability to revisit and revise any part of the sequence provides a powerful new tool for robust and controllable generation, transcending the immutable, left-to-right approach of AR LLMs (Han et al., 2023; Varma et al., 2025; Wang et al., 2025c). Recent large-scale systems extend DLLMs to multimodal and instruction-following settings (Yang et al., 2025; Zhu et al., 2025a; You et al., 2025), showing the paradigm's versatility. These DLLMs fall into two primary families: discrete (based on mask tokens) and continuous (operating on latent spaces), with discrete models currently dominating in performance. For our work, all compared DLLMs are of the discrete type.

Despite rapid progress, systematic, compute-matched comparisons between DLLMs and strong AR baselines on multi-step reasoning remain scarce. This paper offers a timely analysis: a controlled, side-by-side study of efficiency, accuracy, and failure modes that clarifies AR- versus diffusion-style reasoning paradigms, and distills implications for future research.

## 3 PRELIMINARIES

We view reasoning as *trajectory formation* conditioned on an input $x$ (prompt) that yields an output sequence $y = (y_1, \ldots, y_T)$. AR LLMs directly generate discrete tokens; DLLMs refine a continuous latent trajectory $(z_1, \ldots, z_K)$ and then discretize to tokens.

## 3.1 AUTOREGRESSIVE LLMS

An AR LLM factorizes the conditional distribution as:

$$p_\theta(y \mid x) = \prod_{t=1}^{T} p_\theta(y_t \mid x, y_{<t}), \tag{1}$$

and is trained by minimizing token-level cross-entropy,

$$\mathcal{L}_{\text{CE}}(\theta) = -\mathbb{E}_{(x,y)} \sum_{t=1}^{T} \log p_\theta(y_t \mid x, y_{<t}). \tag{2}$$

Left-to-right decoding (*e.g.*, greedy, beam, or stochastic sampling, optionally with post-training supplements, such as chain-of-thought prompting (Wei et al., 2022; Zhou et al., 2023; Kojima et al., 2022; Suzgun et al., 2023; Fu et al., 2023), test-time scaling (Wang et al., 2023b; Yao et al., 2023;

Cobbe et al., 2021; Wu et al., 2024b;a; Chen et al., 2024; Wang et al., 2025d; Knappe et al., 2024; Yu et al., 2025; Kumar et al., 2025; Zhang et al., 2024b; Joshi et al., 2025)) is *commitment-based*: once a token is emitted, later steps condition on it, which yields scalability and streamability while making early mistakes hard to repair without external search.

For cost intuition, if $L$ denotes the number of generated tokens (including any CoT-like attempts), $S$ the number of independent samples, $B$ the beam width, and $V$ the number of verifier/reranking passes, a rough per-query accounting is:

$$\text{Cost}_{\text{AR}} \approx c_{\text{tok}} L (S + B + V), \tag{3}$$

with $c_{\text{tok}}$ the per-token forward cost. Test-time scaling aligns naturally with this decomposition. The most well-known attempts are self-consistency (Wang et al., 2023b), which increases $S$ and beam search, which enlarges $B$ (Vijayakumar et al., 2018). Note that in our study, we exclude the results that AR LLMs trained with reinforcement learning (RL) for fair comparisons, as diffusion-based LLMs remain in an early stage (Christiano et al., 2017; Ouyang et al., 2022; Zelikman et al., 2022; Hao et al., 2025) and RL approaches for them are not yet sufficiently developed.

## 3.2 DIFFUSION-BASED LLMS

DLLMs are another visible path to achieve the intelligence exhibited by AR LLMs (Nie et al., 2025; Gong et al., 2025b): A forward masking process gradually replaces tokens in the original sequence $x_0$ with special [MASK] tokens (*i.e.*, M), producing a partially masked sequence $x_t$ via:

$$\mathcal{L}(\theta) \triangleq -\mathbb{E}_{t, x_0, x_t} \left[ \frac{1}{t} \sum_{i=1}^{L} \mathbf{1} \left[ x_t^i = \text{M} \right] \log p_\theta \left( x_0^i \mid x_t \right) \right], \tag{4}$$

Its text generation follows a reverse-time refinement $z_K \rightarrow z_0$ before a discretization map $y = g(z_0; x)$ yields tokens. Because refinement edits the whole latent, DLLM supports *global reorganization* and *backtracking* prior to discretization. Its inference cost scales primarily with the number of denoising steps $K$ (and optional restarts $R$),

$$\text{Cost}_{\text{Diff}} \approx c_{\text{step}} K R + c_{\text{disc}}, \tag{5}$$

where $c_{\text{step}}$ is the per-step cost and $c_{\text{disc}}$ the discretization overhead. In our study, we vary schedule length $K$ and guidance under matched computational budgets (see §5) so that any observed gains reflect the pathway rather than the extra compute.

Notice the difference between the cost of AR LLMs (see Eq. 3) and DLLMs (see Eq. 5), we hold constant datasets, prompts, model size and evaluation scripts to set up a fair benchmark. To test their stability, we sweep AR knobs (*i.e.*, beam width, self-consistency samples) and diffusion knobs (*i.e.*, schedule length $K$, guidance scale) under controlled settings.

## 4 UNDER MATCHED SCALE, DLLMS OUTPERFORM AR LLMS ON MOST REASONING TASKS BUT INCUR MUCH HIGHER COMPUTATIONAL LATENCY

Our investigation starts with experimental analysis to address **RQ1**, assessing differences in reasoning performance between AR and DLLMs. We outline the baselines and datasets used in this study, which also apply to address RQ2 (see §5).

**Baselines.** We evaluate AR and DLLMs of comparable scale (*i.e.*, 7∼8B parameters). AR baselines include Llama 3.1 8B (Touvron et al., 2023), Mistral 8B (Jiang et al., 2023), and DeepSeek 7B (Dai et al., 2024). DLLM baselines are Dream 7B (Ye et al., 2025b) and LLaDA 8B (Gong et al., 2025a).

**Datasets.** Our evaluation covers 13 benchmarks, which span a total of 20 tasks across four representative categories: quantitative reasoning, logical consistency, semantic entailment, and commonsense QA, following prior surveys (Yu et al., 2024; Sprague et al., 2025). Among them, Minerva Math consists of 7 sub-tasks and ANLI is composed of 3 rounds, which we count separately as tasks.

For *mathematical reasoning*, we use GSM8K (Cobbe et al., 2021), MathQA (Amini et al., 2019), and Minerva Math (Hendrycks et al., 2021). For *natural language inference*, we include ANLI (Nie et al., 2020), MNLI, and RTE (Williams et al., 2018). For *logical reasoning*, we test on LSAT-LR and LogiQA-en (Liu et al., 2020). For *commonsense QA*, we consider COPA (Roemmele et al., 2011), PIQA (Bisk et al., 2020), OpenBookQA (Mihaylov et al., 2018), and HellaSwag (Zellers et al., 2020). All datasets are accessed via Hugging Face.

**Evaluation Metrics.** We use exact match accuracy for GSM8K, MathQA, and Minerva Math; classification accuracy for ANLI, MNLI, and RTE; and multiple-choice accuracy for LSAT-LR, LogiQA-en, COPA, PIQA, Hellaswag, and OpenBookQA. More details are shown in Appendix §S1.

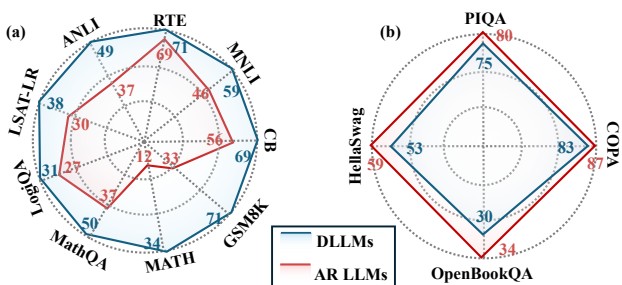

Figure 2: (a) **DLLMs** achieve higher accuracy on **math, logic, and NLI benchmarks**. (b) **AR LLMs** achieve higher accuracy on **commonsense QA**. Here, **MATH** refers to the Minerva MATH dataset (Hendrycks et al., 2021).

**Benchmark Results.** DLLMs achieve consistent advantages on benchmarks that demand stronger global consistency reasoning. Their iterative refinement helps in complex reasoning (see §2.2), whereas AR LLMs follow a fast left-to-right pathway, which remains competitive on simpler commonsense tasks. This motivates the deeper mechanism analyses in §5. In Fig. 2, we present the overall trends of AR LLMs and DLLMs, reporting average performance derived from AR LLMs and three DLLMs evaluated over 20 tasks. As seen, DLLMs averagely outperform AR counterparts on the majority of benchmarks, achieving higher accuracy on **16 of 20** tasks (*e.g.*, GSM8K, MathQA, LogiQA, ANLI, MNLI). AR LLMs remain competitive on a smaller subset of commonsense QA tasks (*e.g.*, PIQA, COPA, OpenBookQA, and Hellaswag). This demonstrates a clear performance gap in favor of diffusion decoding under matched model scales. On average, DLLMs improve accuracy by ∼12 points over comparable AR LLMs under matched scale. For completeness, Appendix §S2 provides the detailed results of *each individual* AR LLM and DLLM, where the trend remains consistent.

**Computational Efficiency Results.** We further report FLOPs/token, average output length, throughput, and latency in Table 1. As seen, DLLMs require higher computational cost per token than AR baselines (*e.g.*, 16,120 GFLOPs/token for Dream-7B vs. 15 GFLOPs/token for Llama-3.1-8B), and also produce slightly

Table 1: **Computational efficiency** comparison on GSM8K. ♪ denotes DLLMs (same for Table 2∼4).

| Model | FLOPs/Token (GFLOPs) | Avg. Tokens per Answer | Throughput (samples/s) | Latency (vs. Llama) |
|---|---|---|---|---|
| Dream-7B ♪ | 16,120[1] | 55.18 | 0.101 | 25.9× |
| LLaDA-8B ♪ | 19,590[1] | 63.37 | 0.064 | 27.4× |
| Llama-3.1-8B | 15.0 | 53.05 | 2.49 | 1.0× |
| Mistral-7B | 14.2 | 57.44 | 1.85 | 0.5× |
| DeepSeek-7B | 14.0 | 54.77 | 1.92 | 0.4× |

longer outputs on average. Throughput is correspondingly low, with DLLMs generating only ∼0.1 samples/s compared to 1.8–2.5 samples/s for AR LLMs. Latency comparisons further confirm this gap: diffusion decoding is consistently slower, often by 10×–200× depending on the number of denoising steps (*e.g.*, on GSM8K, Dream-7B is 25.9× slower and LLaDA-8B is 27.4× slower). All these trends stem from the DLLMs' iterative refinement process, which demands many denoising steps per output. AR, on the other hand, is highly efficient, producing sequences in a single forward pass with much higher throughput. This highlights the key bottleneck of current DLLMs: while they improve reasoning accuracy, their practical deployment is constrained by computational efficiency. For completeness, results on other tasks are shown in Appendix §S5.

**Chain-of-thought (CoT) Prompting Results.** Among post-training methods, CoT prompting is widely adopted to improve reasoning in AR LLMs (Wei et al., 2022). We thus naturally extend CoT to DLLMs, and evaluate CoT+AR LLMs' and CoT+DLLMs' performance, respectively.

Table 2: **Effect of CoT prompting.**

| Model | GSM8K | | MATH | |
|---|---|---|---|---|
| | w/o CoT | w/ CoT | w/o CoT | w/ CoT |
| LLaDA-8B ♪ | 70.7 | 70.4 | 31.5 | 28.0 |
| Llama-3.1-8B | 49.9 | 58.8 | 18.2 | 20.1 |

Following prior work showing that mathematical reasoning benchmarks most clearly reveal the benefits of CoT prompting (Sprague et al., 2025), we evaluate their performance on the GSM8K(Cobbe et al., 2021) and Minerva Math (Hendrycks et al., 2021) , which consists of 7 sub-tasks. As shown in Table 2, LLaDA (DLLM) shows a slight decrease in accuracy on both GSM8K and MATH, while Llama (AR) improves substantially on both benchmarks (*e.g.*, +9 on GSM8K, +2 on MATH). These results demonstrate that CoT prompting yields substantially greater benefits for Llama than for LLaDA. These results highlight that DLLMs remain stronger in absolute terms, but AR LLMs benefit far more from CoT prompting. It suggests that DLLMs already rely on iterative refinement to

enforce intermediate consistency, so additional explicit reasoning offers limited gains. By contrast, AR LLMs profit markedly from such external scaffolding, which helps mitigate their vulnerability to early commitment errors. More detailed CoT results are provided in Appendix S6.

## 5    DLLMS OUTPERFORM AR LLMS ON GLOBAL-CONSTRAINT TASKS, UNDER NOISY PROMPTS, AND AT MATCHED COMPUTATIONAL BUDGET

After establishing the overall performance study (RQ1), we turn to **RQ2**, analyzing when DLLMs outperform AR LLMs and further ask why. We first define the task taxonomy based on our evaluated datasets, and further observe three consistent patterns.

**Tasks Taxonomy.** In our study, we categorize tasks into two types: single-constraint and multi-constraint tasks. ***Single-constraint tasks*** generally require a single dominant decision guided by local cues. For example, the task is to choose the most plausible option in commonsense QA benchmarks such as PIQA and COPA (see Appendix §S3). ***Multi-constraint tasks***, on the other hand, require satisfying multiple interdependent constraints *jointly*. For instance, math tasks such as GSM8K and MATH demand consistency across intermediate variables, where each step's result must align with subsequent operations. Similarly, logical reasoning in LogiQA requires identifying which premises are relevant to the question and reasoning about their interrelations, ensuring consistency across multiple pieces of evidence. Natural language inference tasks such as ANLI further demand global semantic alignment between premise and hypothesis, covering all relevant details rather than relying on surface

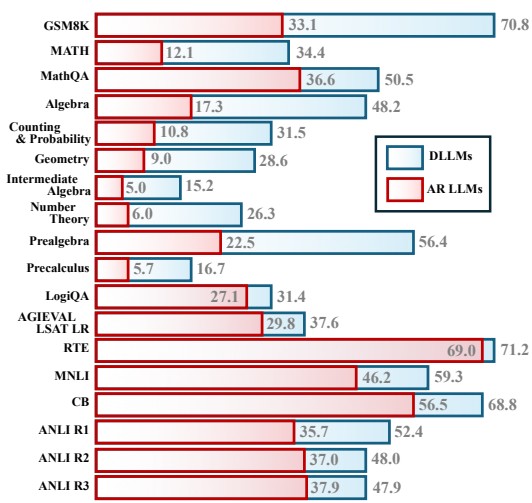

Figure 3: **DLLMs outperforms AR LLMs on tasks requiring muti-constraints**, including math reasoning, multi-premise logic, and NLI.

overlap. In our study, while AR LLMs prefer single-constraint tasks (see Appendix S3), DLLMs achieve noticeable advantages in multi-constraint tasks (see **Finding I**).

**Finding I: DLLMs win on global, multi-constraint consistency.** DLLMs achieve superior accuracy on tasks requiring *muti-constraint satisfaction* and *global coherence*, such as math with intermediate bookkeeping, multi-premise logic, and sentence-pair entailment. DLLMs' iterative denoising updates the *entire sequence in parallel*, enabling repeated checks and repairs of cross-token dependencies. This allows the model to satisfy multiple constraints simultaneously (*e.g.*, content, numeric validity, logical structure), instead of committing early to a single trajectory. Concretely, we examine the strengths of DLLM reasoning compared to AR LLMs on three representative multi-constraint tasks (*i.e.*, mathematical reasoning, natural language inference, and logical reasoning).

As presented in Fig. 4, the example DLLM (*i.e.*, LLaDA) demonstrates its effectiveness in complying with multiple constraints where AR LLM (*i.e.*, Llama) fails to do so. **I.** On GSM8K, a mathematical reasoning benchmark of math word problems where multiple quantitative constraints are given, correct solutions require satisfying all given constraints simultaneously. In Fig. 4(a), AR LLM Llama neglects constraints 1 and 2, resulting in a fatal error that incorrectly calculates the cost of supplies for 20 candles, which finally leads to a completely wrong solution. In contrast, DLLM LLaDA successfully captures all the key quantitative constraints and produces a final correct solution. **II.** On ANLI, a natural language inference benchmark where models must respect semantic and lexical constraints as well as factual boundaries established by the premise to make logically consistent inferences about the hypothesis, we observe similar patterns. The example in Fig. 4(b) demonstrates that Llama fails to comply with the lexical constraints, while LLaDA successfully detects the lexical inconsistency and predicts the correct answer. **III.** On a more challenging logical reasoning benchmark LogiQA, as shown in Fig. 4(c), Llama fails to satisfy spatial constraint

---

[1]FLOPs data for Dream-7B and LLaDA-8B are reported from (Liu et al., 2025).

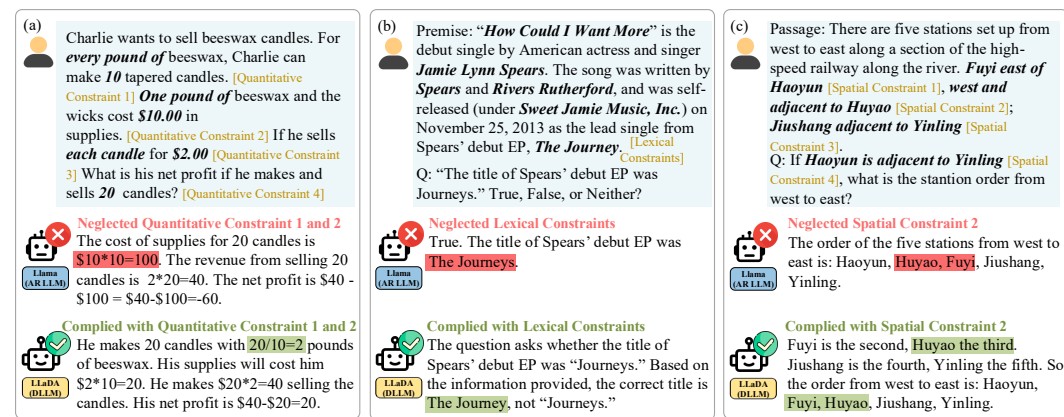

**Figure 4: Case analysis of AR LLM and DLLM on multi-constraint reasoning tasks.** Examples show AR LLM (Llama) failures due to constraint neglect on (a) GSM8K mathematical reasoning, (b) ANLI natural language inference, and (c) LogiQA logical reasoning. The colors indicate whether one or more constraints are neglected or complied with during the reasoning process.

2, leading to incorrect logical reasoning. In contrast, LLaDA captures all the logical cues and constraints, demonstrating superior constraint adherence. These observations highlight that DLLMs' parallel decoding mechanism provides robustness against constraint neglecting and multi-constraint satisfaction failures that AR LLMs suffer from. Additional cases are provided in Appendix S4.

**Finding II: DLLMs are more robust to prompt noise.** Under prefix and suffix perturbations, AR decoding couples tightly to corrupted input, causing small changes to cascade through the entire sequence. DLLMs, in contrast, *gradually correct* errors via multiple denoising steps under full context, enabling partial or strong recovery depending on noise severity.

Table 3: Accuracies **under noise perturbations.**

| Model | Clean | Prefix | Suffix |
|---|---|---|---|
| LLaDA ♪ | 100 | 68 | 50 |
| Llama | 100 | 31 | 36 |

To evaluate robustness, we construct a GSM8K subset where both Llama (AR LLM) and LLaDA (DLLM) solve all problems correctly under clean prompts, and then inject 15 randomly sampled adversarial tokens either before (*i.e.*, prefix) or after (*i.e.*, suffix) the query (Gan et al., 2024; Qiang et al., 2024; Anantheswaran et al., 2024). As shown in Table 3,

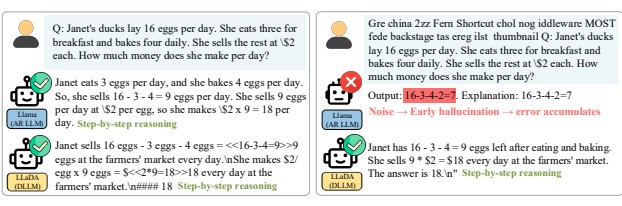

Figure 5: Adversarial noise triggers **AR LLMs error accumulation**, while **DLLM remains stable**

results are evaluated on a GSM8K subset where both models are correct under clean prompts.

Under prefix and suffix noise attacks, Llama's accuracy collapses to 34%, while LLaDA maintains substantially higher robustness at 50%. This gap reflects the distinct error behaviors of the two paradigms under perturbations. DLLMs, though still affected by perturbations, can partially recover and preserve reasoning consistency, whereas AR LLMs are more tightly coupled to the injected noise. Under clean prompts, both models produce step-by-step arithmetic reasoning and correct answers. After injection, Llama often outputs an anomalously large spurious number as its first step and then rationalizes it, revealing irreversible error accumulation. By contrast, LLaDA maintains stable intermediate computations and recovers the correct answer (see Fig. 5). This suggests that iterative denoising mitigates perturbations and stabilizes the reasoning trajectory.

Overall, these results connect task demands with decoding dynamics: DLLM's parallel refinement favors tasks needing global consistency and robustness to perturbations, while AR's causal composition favors tasks reducible to a single left-to-right reasoning chain.

**Finding III: DLLMs maintain their advantages under matched computational budget.**

We further examine computational overhead using inference FLOPs. DLLMs inherently require multiple denoising iterations during decoding, resulting in higher computational demands compared to AR LLMs. To isolate the effects of model architecture from computational cost, we adopt the

compute-matching setting from Wu et al. (2024a). In this controlled setting, AR LLM inference is executed multiple times via test-time scaling (Wang et al., 2023b; Wu et al., 2024a; Wang et al., 2025d), while LLaDA-8B (DLLM) are equipped with KV cache (Liu et al., 2025) matching the caching mechanism in Llama-3.1-8B (AR LLMs) to ensure a fair comparison. As shown in Table 4, we run $n = 140$ and $n = 560$ samples for Llama, aligned to 256 denoising steps and 512 steps with generate length 512 for LLaDA, respectively. Even under these matched-compute settings, LLaDA continue to consistently outperform Llama (*i.e.*, 74 vs. 68% at $1.0\times$, and 78% vs. 69% at $4.0\times$).

LLaDA delivers consistently higher accuracy than Llama. The advantage persists across budgets, showing that additional AR test-time sampling cannot close the gap. This finding underscores that DLLMs' strengths come from their refinement mechanism rather than from extra computational resources. Due to computational resource limits, these matched-compute comparisons were conducted on a 100-example subset of GSM8K. We also tested alter-

Table 4: Accuracy under **matched compute resources**. † LLaDA results are reported with KV cache.

| Configuration | FLOPs | Acc. (%) |
|---|---|---|
| LLaDA-8B ♪†, 256 steps | $1.0\times$ | **74** |
| Llama-3.1-8B, $n = 140$ | $1.0\times$ | 68 |
| LLaDA-8B ♪†, 512 steps + 512 gen | $4.0\times$ | **78** |
| Llama-3.1-8B, $n = 560$ | $4.0\times$ | 69 |

native test-time scaling methods, such as beam search, but found that AR LLMs still underperform DLLMs. More details and results are provided in Appendix §S7.

# 6 DLLMs Require Structured Discretization, Long Schedules, and Balanced Hyperparameters

Table 5: **Ablation studies** for DLLM (LLaDA) on GSM8K. Unless otherwise specified, the default setting uses 256 denoising steps, generates length 256, block length 256, low-confidence masking, CFG 0.0, and temperature 0.0. The adopted designs are marked in red.

| (a) Discretization | | (b) Guidance | | (c) Schedule | | (d) Generate | | (e) Block | | (f) Temperature | |
|---|---|---|---|---|---|---|---|---|---|---|---|
| Scheme | Acc. (%) | Strength | Acc. (%) | Steps | Acc. (%) | Length | Acc. (%) | Block Size | Acc. (%) | Temp | Acc. (%) |
| Random | 15.6 | **CFG 0.0** | **70.7** | 64 steps | 40.5 | **256** | **70.7** | **256** | **70.7** | 0.0 | 70.7 |
| **LowConf** | **70.7** | CFG 0.5 | 70.2 | 128 steps | 61.2 | 512 | 69.3 | 128 | 69.0 | **0.2** | **71.1** |
| — | — | CFG 1.0 | 65.5 | **256 steps** | **70.7** | 1024 | 47.2 | 64 | 68.4 | 0.7 | 68.9 |
| — | — | CFG 1.5 | 61.3 | — | — | — | — | — | — | 1.0 | 68.1 |

We then answer **RQ3**, searching the hyperparameter trends for DLLMs to maximize their reasoning performance. Specifically, we conduct systematic ablations on GSM8K, varying discretization schemes, guidance strengths, denoising schedules, temperature, and sampling strategy, where we recognize that these factors directly control how DLLM refines sequences.

**Discretization Strategy.** Diffusion decoding refines sequences by repeatedly re-sampling a subset of tokens at each step. There are two common schemes: (i) *random masking*, where tokens are re-sampled uniformly at random regardless of model uncertainty, and (ii) *low-confidence masking*, where only tokens with the lowest predicted confidence are re-sampled (Li et al., 2022; Sahoo et al., 2024). Shown in Table 5(a), the low-confidence strategy consistently achieves strong accuracy (*e.g.*, 70% on GSM8K), whereas random masking collapses performance to 16%. This pattern demonstrates that DLLMs depend on *structured refinement*. Focusing updates on uncertain tokens enables the model to progressively repair errors without disturbing already correct tokens. In contrast, unstructured randomness introduces noise into stable regions, disrupting global coherence and breaking the refinement trajectory.

**Classifier-free Guidance (CFG).** CFG (Ho & Salimans, 2021; Li et al., 2022; Chung et al., 2025; Han et al., 2024) scales the conditional score against the unconditional one, controlling how strongly the model follows the prompt versus exploring alternative completions. As shown in Table 5(b), moderate guidance (*i.e.*, $0.0 - 0.5$) preserves accuracy at 70%, while stronger guidance (*i.e.*, $1.0 - 1.5$) reduces accuracy to $66-61\%$. *Why does strong guidance hurt?* High guidance amplifies already confident tokens overly, which in turn suppresses diversity and the model's capacity for self-correction. As a result, early errors are reinforced instead of revised. Moderate guidance, in contrast, preserves a balance between following the prompt and leaving flexibility to repair mistakes, which is sufficient for DLLMs since its iterative process already enforces global consistency.

**Denoising Schedule Length.** A key advantage of DLLMs lies in their ability to correct errors through iterative denoising (Ye et al., 2025a; Zhao et al., 2025). As shown in Fig. 6, performance improves as the number of iteration steps increases.

At 64 steps, multiple mistakes remain, and the reasoning process is unstable to DLLM. At 128 steps, most errors are eliminated, and the overall structure becomes gradually coherent. At 256 steps, all errors are corrected, and the solution converges to the correct answer. This progression shows

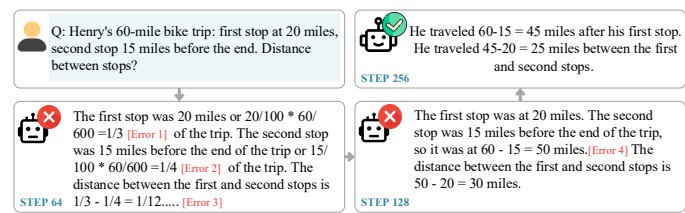

Figure 6: **Error correction** in DLLMs as denoising progresses.

that DLLMs refine incomplete or faulty reasoning into coherent solutions through repeated denoising cycles. The number of steps determines the refinement depth. Too few leave many errors unresolved, while additional iterations gradually enforce global consistency. As reported in Table 5(c), accuracy increases from 40.5% at 64 steps to 61.2% at 128 steps, and reaches 70.7% at 256 steps.

**Generate Length.** Increasing the generate length $L_g$ produces longer outputs but splits decoding into more blocks, thereby reducing refinement steps per block (Arriola et al., 2025). While the total denoising iterations remain constant, each block receives fewer updates, weakening local correction. As shown in Table 5(d), accuracy peaks at $L_g$=256, whereas longer lengths gradually cause reasoning drift and steadily reduce overall coherence of the solution process.

**Block Length.** The block length specifies the span of each local update. Table 5(e) shows that DLLM's accuracy reaches the highest when the block length is set to 256. In this case, the generate and block lengths are identical, allowing the model to refresh the entire sequence per iteration. Under this setting, DLLM avoids semantic fragmentation and ensures that local updates remain consistent with the global context. Reducing the block length (*e.g.*, 128 or 64) fragments the updates, which weakens coordination across reasoning steps and results in lower accuracies.

**Temperature.** DLLM's temperature regulates randomness in token sampling. At $T$=0, decoding is deterministic. At very high $T$ (*e.g.*, 0.7∼1.0), outputs become unstable and reasoning drifts (Finlayson et al., 2024; Renze, 2024; Shih et al., 2023; Chang et al., 2023; Zhang et al., 2024a). Shown in Table 5(e), a moderate value around 0.2 achieves best balance. It enables exploration of alternatives while maintaining coherence, allowing DLLM to fix local errors without losing track.

Overall, the results indicate that DLLMs' effectiveness is strongly shaped by parameter choices. It is important to adopt principled configurations to achieve reliable performance.

# 7 DISCUSSION AND CONCLUSION

We distill our findings into immediate guidance for practitioners. Our analysis suggests that AR LLMs and DLLMs resemble computational analogues of *fast* and *slow* thinking, respectively: AR LLMs generate rapidly via greedy, token-by-token decoding (*i.e.*, System 1–like efficiency), whereas DLLMs iteratively refine a global representation (*i.e.*, System 2–like deliberation). We use this lens to analyze each paradigm's characteristic strengths and liabilities in reasoning.

The primary engineering challenge for DLLMs is computational cost. Iterative refinement is powerful but raises FLOPs and latency, limiting suitability under strict service-level constraints in production and large-scale serving contexts. Promising mitigation include step distillation (Chen et al., 2025; Xie et al., 2024; Salimans et al., 2024; Zhou et al., 2024b; Zhu et al., 2025b; Ho & Salimans, 2022), rectified-flow and related training (Lee et al., 2024; Zhu et al., 2024; Lipman et al., 2023; Wang et al., 2025a), and consistency training (Song et al., 2023; Song & Dhariwal, 2024; Dao et al., 2025). In particular, learned step-size schedules and adaptive stopping criteria should be prioritized to tighten the quality–latency trade-off without eroding DLLMs' capacity for global correction.

Our evaluation also reveals a clear crossover regime. For tasks with short causal chains and tight latency budgets, AR LLMs tend to dominate; additional DLLM iterations add unnecessary deliberation and may even depress accuracy. Conversely, on problems with multi-variable dependencies and long-range coherence requirements, DLLMs excel by progressively enforcing global consistency and reducing internal contradictions. These results naturally motivate hybrid designs: use a two-stage pipeline where an AR LLM produces a draft and a DLLM refines it to enforce constraints and resolve inconsistencies, or adopt dynamic routing that estimates task difficulty online and selects the appropriate pathway, allocating compute where it yields the highest return.

ETHICS STATEMENT

We conform to the ICLR Code of Ethics and further show the consent to our work below. All datasets used in this study are publicly available and released under permissive licenses (see Appendix §S9), and all the models are publicly available (see Appendix §S9 for Asset License and Consent). We would like to state that the contents in the dataset do NOT represent our views or opinions and our paper does not involve crowdsourcing or research with human subjects.

REPRODUCIBILITY STATEMENT

All experiments in this paper are evaluation-only. Our implementation is based on PyTorch (Paszke et al., 2019) and runs on NVIDIA A100-40GB GPUs. We evaluate publicly available models on publicly available datasets (see Appendix §S9 for details). We provide exact dataset and evaluation metrics Appendix §S1, so that our reported results can be reproduced. Our evaluation scripts will be released upon acceptance.

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

- §S1 introduces the **datasets**, covering statistics, task categories, and evaluation metrics.
- §S2 reports **additional results**, and shows radar charts that compare models across benchmarks.
- §S3 presents **failure cases in single-constraint tasks** and explains why AR LLMs outperform DLLMs in these settings.
- §S4 presents **failure cases in multi-constraint tasks** and shows that DLLMs outperform AR LLMs when multiple interdependent constraints must be satisfied.
- §S5 provides an **analysis of efficiency scaling**, and reports memory usage and latency as a function of sequence length for AR and DLLMs.
- §S7 provides an **examination of test-time scaling**, and compares AR LLMs with beam search.
- §S9 offers a **summary of licenses and consent**, and lists usage terms for all models and datasets.
- §S10 provides a **discussion of social impact and limitations**, and highlights broader implications and open challenges.
- §S11 provides an **AI disclosure**, and notes that AI assistance was limited to grammar checking.

## S1 DATASETS AND EVALUATION METRICS

### S1.1 DATASETS

We evaluate our models on a broad set of benchmarks spanning mathematical reasoning, natural language inference, logical reasoning, and commonsense question answering. Below we provide dataset descriptions and links for reproducibility.

**Mathematical reasoning.**

- **GSM8K** (Cobbe et al., 2021): a grade school math word problem benchmark. Available at `https://huggingface.co/datasets/openai/gsm8k`.
- **MathQA** (Amini et al., 2019): a collection of math word problems derived from AQuA. Available at `https://huggingface.co/datasets/allenai/math_qa`.
- **Minerva Math** (Hendrycks et al., 2021): a large-scale dataset covering 7 subfields of mathematics. Available at `https://huggingface.co/datasets/EleutherAI/hendrycks_math`.

**Natural language inference.**

- **ANLI** (Nie et al., 2020): an adversarially collected NLI benchmark. Available at `https://huggingface.co/datasets/facebook/anli`.
- **MNLI** (Williams et al., 2018): a broad-coverage NLI dataset. Available at `https://huggingface.co/datasets/nyu-mll/glue`.
- **RTE** (Williams et al., 2018): a textual entailment dataset from the GLUE benchmark. Available at `https://huggingface.co/datasets/nyu-mll/glue`.

**Logical reasoning.**

- **LSAT-LR**: logical reasoning problems from the LSAT exam. Available at `https://huggingface.co/datasets/hails/agieval-lsat-lr`.
- **LogiQA-en** (Liu et al., 2020): an English logical reasoning benchmark. Available at `https://huggingface.co/datasets/hails/agieval-logiqa-en`.

**Commonsense QA.**

- **COPA** (Roemmele et al., 2011): a causal reasoning dataset. Available at `https://huggingface.co/datasets/super_glue/viewer/copa`.

- **PIQA** (Bisk et al., 2020): a physical commonsense reasoning benchmark. Available at `https://huggingface.co/datasets/ybisk/piqa`.
- **OpenBookQA** (Mihaylov et al., 2018): multiple-choice science QA benchmark. Available at `https://huggingface.co/datasets/allenai/openbookqa`.
- **HellaSwag** (Zellers et al., 2020): commonsense completion benchmark with adversarial filtering. Available at `https://huggingface.co/datasets/Rowan/hellaswag`.

### S1.2 EVALUATION METRICS

For **mathematical reasoning tasks** (GSM8K, MathQA, Minerva Math), we use *Exact Match (EM)* accuracy. We consider both strict and flexible EM (the latter allows normalization such as removing commas, units, and checking mathematical equivalence, *e.g.*, $0.5 = 1/2$), as well as Math Verify (MV) for Minerva Math. Following standard practice, we report the highest score among these metrics for each dataset to ensure comparability with prior work.

For **natural language inference tasks** (ANLI, MNLI, RTE), we use *classification accuracy*, *i.e.*, the percentage of samples where the predicted label exactly matches the gold label.

For **multiple-choice QA tasks** (LSAT-LR, LogiQA-en, COPA, PIQA, OpenBookQA, HellaSwag), we use *multiple-choice accuracy*, defined as the proportion of questions where the correct option is selected. We primarily report standard accuracy in the main paper for consistency.

For **multi-subtask datasets** (Minerva Math, ANLI), accuracy is first computed per subtask and then aggregated using sample-weighted averages. This weighting reflects the relative size of each subtask and avoids distortions from smaller subsets.

All metrics are case-insensitive and allow minor formatting variations through regex-based normalization. In practice, this normalization accounts for superficial differences such as spacing, capitalization, or symbol usage.

## S2 ADDITIONAL RESULTS

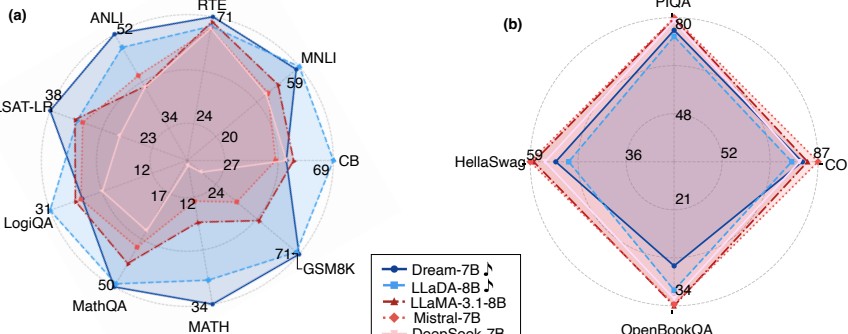

Figure S1: **Radar charts of five models.** (a) Tasks where AR LLMs outperform DLLMs. (b) Tasks where DLLMs outperform AR LLMs.

**AR advantages.** On commonsense QA benchmarks such as PIQA, HellaSwag, COPA, and Open-BookQA, AR LLMs consistently achieve the best performance. For instance, Mistral-7B reaches 80.2% on PIQA and 91.0% on COPA, clearly outperforming diffusion-based models. These results suggest that AR decoding remains more effective on short-context multiple-choice tasks where local token dependencies dominate.

**DLLM advantages.** In contrast, DLLMs (Dream-7B and LLaDA-8B) show strong gains on multi-step reasoning datasets such as GSM8K, MathQA, and RTE. Dream-7B reaches 71.5% on GSM8K

and $50.0\%$ on MathQA, surpassing AR counterparts by large margins. This highlights DLLMs' strength in handling structured reasoning under multiple interdependent constraints.

DLLMs achieve consistent advantages on benchmarks that demand strong global consistency and multi-step reasoning, where iterative refinement enables error correction and coherence. By contrast, AR LLMs remain competitive on short-context commonsense QA, where fast left-to-right decoding is sufficient for capturing single causal links and maintaining reasoning chains.

## S3   FAILURE CASES IN SINGLE-CONSTRAINT TASKS

**Finding: Autoregressive models wins on single-constraint reasoning.** When tasks are driven by *one dominant causal or procedural relation*, as in commonsense and procedural QA, AR LLMs perform better. Left-to-right decoding $(p(x) = \prod_i p(x_i \,|\, x_{<i}))$ constructs a stable and interpretable chain, where each step relies directly on the previous one. This concentrates probability mass on the "next sensible step," reducing uncertainty and avoiding iterative dilution. Such single-pass reasoning aligns well with single-constraint tasks. As shown in Fig. S1(a), AR LLMs achieve higher accuracy on PIQA, COPA, OpenBookQA and Hellaswag.

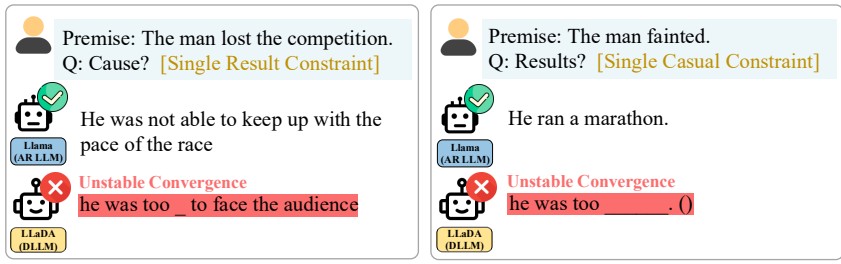

Figure S2: **Autoregression outperforms diffusion on single-constraint reasoning.** Examples from commonsense reasoning tasks show that Llama (AR) provides coherent causal links, while LLaDA (DLLM) produces incomplete or redundant reasoning, illustrating difficulties in converging on discrete cause–effect constraints.

While DLLMs excel at multi-constraint reasoning, they often underperform AR LLMs on single-constraint tasks. Figure S2 shows two representative examples from commonsense reasoning benchmarks. Autoregressive models generate tokens causally, which enforces strong local coherence. This makes them effective when solving problems reducible to a single causal link (*e.g.*, identifying a direct cause or a single plausible outcome). As seen in the examples, Llama produces fluent and consistent explanations such as "he was not able to keep up with the pace of the race." In contrast, DLLMs rely on bidirectional denoising, which weakens causal flow and may yield incomplete fragments rather than converging on the correct relation.

As a result, DLLMs sometimes generate incomplete fragments ("he was too _ to face the audience"), failing to capture the intended causal relation. This reflects a general weakness of DLLMs in tasks where precision hinges on a single constraint.

## S4   ADDITIONAL FAILURE CASE ANALYSES IN MUTI-CONSTRAINT TASKS

We provide further failure case analyses to complement the main text. Figure S3 illustrates four multi-constraint tasks where diffusion consistently outperforms autoregression.

In Fig. S3(a), Llama (AR LLM) ignores constraints 1 and 2, producing inconsistent arithmetic, while LLaDA (DLLM) satisfies all quantitative conditions. In Fig. S3(b), Llama fails to account for the temporal relation (*2011 vs. 2008*), whereas LLaDA preserves lexical consistency. In Fig. S3(c), Llama omits the total-sum requirement in a story aggregation task, but LLaDA produces the correct outcome. In Fig. S3(d), Llama misreads the average constraint as a daily increment, leading to compounding errors, while LLaDA maintains global coherence.

These additional cases further demonstrate DLLMs' ability to maintain global consistency under multi-constraint reasoning.

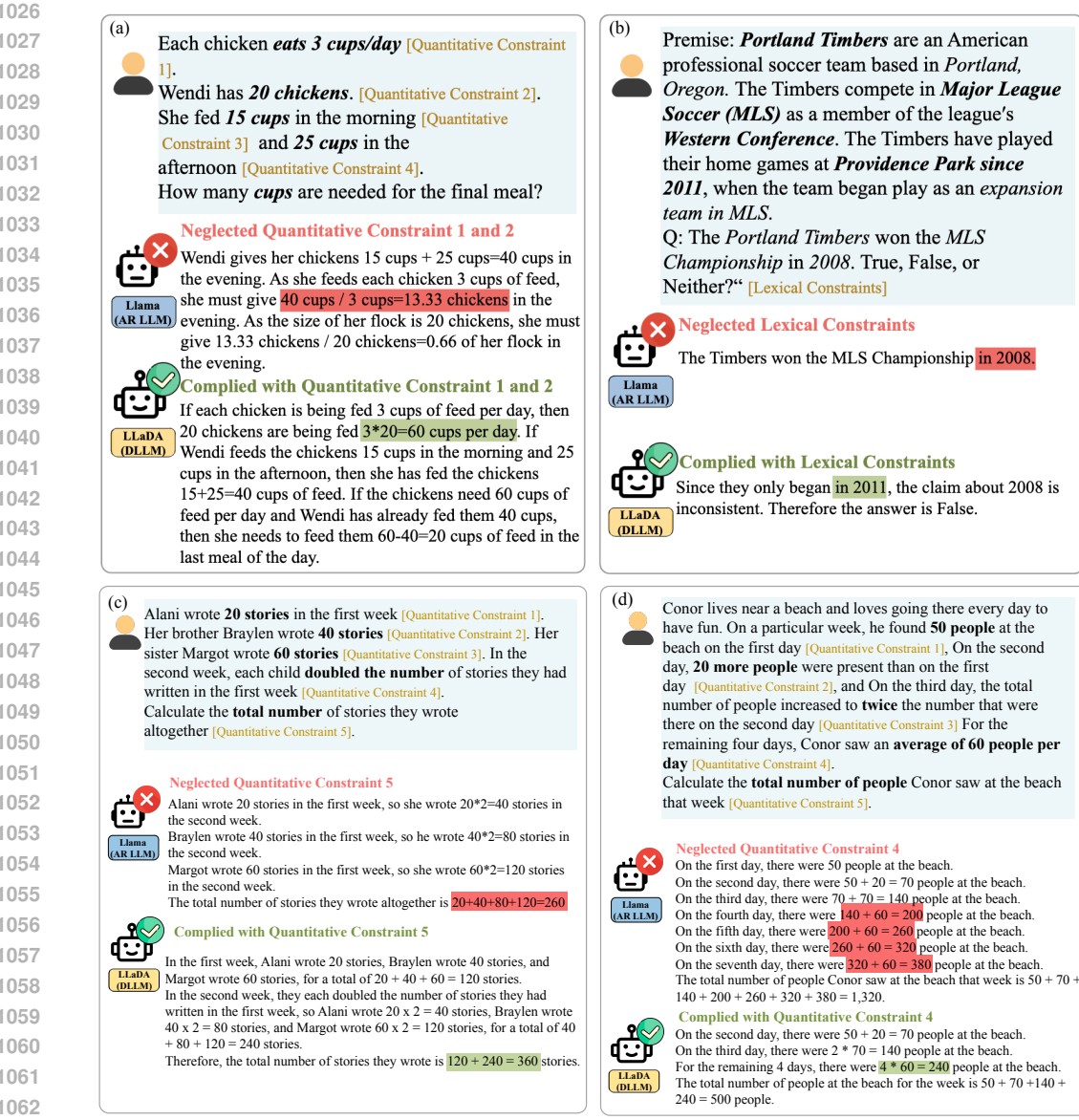

Figure S3: **DLLMs win on multi-constraint reasoning.**

## S5    EFFICIENCY

A critical dimension differentiating reasoning paradigms is computational efficiency, particularly latency and memory consumption, which directly impact practical deployment. We conducted stress tests on both LLaDA (DLLM) and Llama (AR LLM) across varying sequence lengths.

As shown in Table S1, Llama's latency remains nearly constant (5–6s) across prompt lengths from 32 to 2048 tokens, indicating near-linear scalability. In contrast, LLaDA exhibits sharp latency growth: from 11.2s at 32 tokens to 62.8s at 2048 tokens. This divergence reflects the fundamental difference between paradigms: AR decoding processes tokens sequentially with stable cost per step, while diffusion-based decoding incurs iterative refinement, with costs rising sharply as context length grows. This shows DLLMs face challenges in handling long contexts efficiently.

This analysis confirms a fundamental trade-off: the reasoning strengths of DLLMs come at a significant cost in terms of computational efficiency, whereas AR LLM offer a much more scalable and resource-friendly solution, making them better suited for real-time or resource-constrained applications. Deploying diffusion decoding therefore requires careful attention to latency despite its accuracy and robustness gains.

Table S1: **Latency as a function of prompt length.** Llama's cost remains stable, while LLaDA grows sharply with longer prompts. Latency is measured in seconds.

| Prompt Length | Llama-3.1-8B | LLaDA-8B |
|:---:|:---:|:---:|
| 32 | 5.5 | 11.2 |
| 64 | 5.5 | 11.5 |
| 128 | 5.6 | 12.1 |
| 256 | 5.6 | 15.3 |
| 512 | 5.7 | 21.1 |
| 1024 | 5.7 | 35.9 |
| 2048 | 5.7 | 62.8 |

This analysis confirms a fundamental trade-off: the reasoning strengths of DLLMs come at a significant cost in terms of computational efficiency, whereas AR LLM offer a much more scalable and resource-friendly solution, making them better suited for real-time or resource-constrained applications. Deploying diffusion decoding therefore requires careful attention to latency despite its accuracy and robustness gains.

Table S2 compares end-to-end inference latency across benchmarks, normalized to Llama-3.1-8B. We find that diffusion decoding is consistently slower, often by $4\times$–$43\times$, depending on the number of denoising steps. For instance, on GSM8K, Dream-7B is $25.9\times$ slower, and LLaDA-8B is $27.4\times$ slower. This highlights the key bottleneck of current diffusion LLMs: while they improve reasoning accuracy, their practical deployment is limited by inference speed (Li et al., 2022).

Table S2: **Latency comparison** across different tasks relative to Llama-3.1-8B.

| Models | DLLM | | AR LLM | | |
|:---:|:---:|:---:|:---:|:---:|:---:|
| | Dream | LLaDA | Llama | Mistral | DeepSeek |
| Parameter | 7B | 8B | 8B | 7B | 7B |
| Mathematical Reasoning | | | | | |
| GSM8K | 25.9× | 27.4× | 1.0× | 0.5× | 0.4× |
| MathQA | 15.5× | 17.4× | 1.0× | 1.0× | 1.0× |
| Minerva Math | 15.6× | 17.6× | 1.0× | 0.7× | 0.7× |
| Natural Language Inference/QA | | | | | |
| MNLI | 3.3× | 3.9× | 1.0× | 0.9× | 1.1× |
| RTE | 9.7× | 9.8× | 1.0× | 1.0× | 1.0× |
| QQP | 5.3× | 5.3× | 1.0× | 1.0× | 1.0× |
| ANLI | 17.9× | 18.0× | 1.0× | 1.1× | 1.1× |
| Logical Reasoning | | | | | |
| LSAT-LR | 38.8× | 43.0× | 1.0× | 1.1× | 1.0× |
| LogiQA-en | 32.0× | 37.2× | 1.0× | 1.1× | 1.0× |
| Commonsense QA Reasoning | | | | | |
| COPA | 5.6× | 4.5× | 1.0× | 1.0× | 0.9× |
| PIQA | 6.9× | 6.4× | 1.0× | 1.1× | 1.0× |
| OpenBookQA | 9.5× | 9.4× | 1.0× | 1.1× | 1.0× |
| HellaSwag | 17.3× | 11.2× | 1.0× | 1.0× | 0.9× |

## S6 CoT RESULTS

Table S3 reports Math Verify (MV) accuracy for LLaDA-8B (DLLM) and Llama-3.1-8B (AR) on the Minerva MATH sub-tasks, both with and without CoT prompting.

For DLLMs, CoT generally fails to provide improvements and sometimes lowers accuracy (*e.g.*, Algebra $41.5 \to 40.0$, Number Theory $22.8 \to 19.8$). In contrast, AR LLMs show modest but consistent gains from CoT in several sub-tasks (*e.g.*, Prealgebra $33.8 \to 38.6$, Geometry $11.3 \to 13.6$). This suggests that iterative refinement in DLLMs already supports multi-step reasoning, so external CoT scaffolding can interfere with their decoding process.

Table S3: **CoT prompting on Minerva Math sub-tasks.** Math Verify (MV) accuracy (%).

| Sub-task | LLaDA-8B (DLLM) | | Llama-3.1-8B (AR LLM) | |
|---|---|---|---|---|
| | w/o CoT | w/ CoT | w/o CoT | w/ CoT |
| Algebra | 41.5 | 40.0 | 27.7 | 29.3 |
| Counting & Prob. | 25.9 | 23.0 | 17.1 | 16.2 |
| Geometry | 19.8 | 18.4 | 11.3 | 13.6 |
| Intermediate Algebra | 9.7 | 9.4 | 6.2 | 7.3 |
| Number Theory | 22.8 | 19.8 | 9.6 | 9.6 |
| Prealgebra | 50.1 | 51.7 | 33.8 | 38.6 |
| Precalculus | 11.4 | 12.8 | 7.5 | 8.4 |

AR LLMs show consistent gains from CoT, while DLLMs exhibit limited or negative response. This suggests that CoT prompting (Wei et al., 2022; Kojima et al., 2022) is more effective for AR LLMs, whereas DLLMs do not benefit under the same setting.

## S7 ADDITIONAL RESULTS ON TEST-TIME SCALING

**Beam search analysis.** We further evaluate the effect of beam search (a common test-time scaling method (Vijayakumar et al., 2018; Koehn & Knowles, 2017)) on Llama (see Table S4). Specifically, we compare greedy decoding ($B = 1$) and beam widths of $B = 2, 4, 8$ on GSM8K. The accuracies are 50.2, 37.4, 38.5, and 37.7, respectively. Despite these adjustments, Llama consistently underperforms compared to diffusion-based models, indicating that simply enlarging the beam does not bridge the gap. This highlights that DLLMs' advantage is not attributable to insufficient search at test time, but rather to their inherent iterative refinement process.

Table S4: **Effect of beam width on Llama vs. LLaDA (GSM8K accuracy).** Beam search does not improve AR performance; all settings remain below DLLMs.

| Beam width ($B$) | Llama (AR LLM) | LLaDA (DLLM) |
|---|---|---|
| 1 (greedy) | 50.2 | |
| 2 | 37.4 | **70.7** |
| 4 | 38.5 | |
| 8 | 37.7 | |

## S8 FUTURE DIRECTIONS

Our study highlights both the strengths and limitations of AR LLMs and DLLMs. Several promising directions emerge for advancing diffusion-based reasoning and hybrid architectures:

**1. Sampling and Efficiency Optimization.** Current DLLMs still depend on dozens or even hundreds of denoising steps, which makes inference slow and resource-intensive. A key research direction is to design more efficient sampling strategies that reduce steps without degrading accuracy. One option is to adopt **adaptive noise schedules**, where the number of refinement steps is dynamically adjusted by token-level uncertainty. Another possibility is **budgeted DLLMs**, in which the model runs under a fixed compute or latency budget and applies early stopping once convergence is detected. In addition, methods such as progressive distillation, caching, or step-sharing across tokens may further accelerate decoding. Together, these approaches aim to close the gap between the reasoning ability of DLLMs and the efficiency required for deployment.

**2. Hybrid AR–Diffusion LLMs Paradigms.** Recent work has already explored various forms of hybridization, such as Block Diffusion (Arriola et al., 2025), demonstrating that AR LLMs and DLLMs can complement each other.

As illustrated in Figure S4, one practical pipeline is a two-stage process:

1. **AR Seeding (Sketch Creation):** An autoregressive model generates an initial, syntactically sound draft, addressing DLLMs's weakness in maintaining tight local dependencies.

2. **Diffusion Refinement (Global Improvement):** The AR output is then treated as a noisy-but-structured input for a DLLMs which iteratively refines the sequence to improve global logic and coherence.

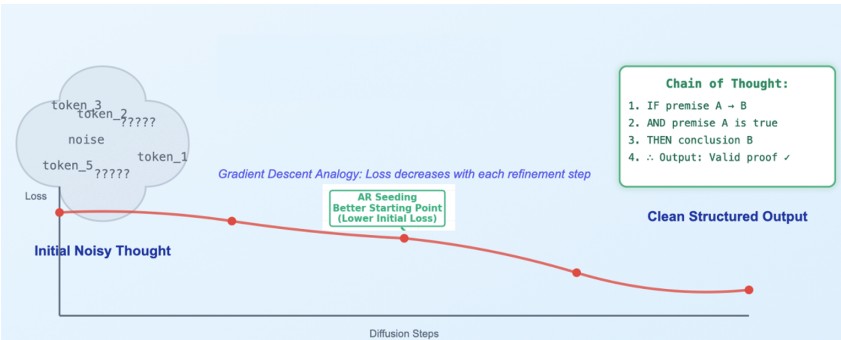

Figure S4: **A Hybrid AR-Diffusion Architecture.** This model first uses an autoregressive transformer to generate a draft or "sketch" of discrete tokens (AR Seeding). This initial output is then fed into a DLLM for iterative refinement, improving global logic and coherence.

This architecture provides a concrete direction for hybrid reasoning systems. Another promising avenue is **task-adaptive routing**, where the decoder dynamically chooses between AR LLMs and DLLMs updates based on the structure of the problem.

**3. Toward Unified Multimodal Reasoning.** Diffusion provides a natural framework for multimodal reasoning because both discrete and continuous signals can be represented within the same denoising process. This opens the possibility of building LLMs that seamlessly integrate text, vision, and other modalities under a shared refinement cycle.

**4. DLLMs as Agents.** The bidirectional context, parallel decoding, and iterative refinement of DLLMs make them promising candidates for agentic applications. Unlike purely autoregressive models, DLLMs can plan and revise their outputs through repeated refinement, which resembles the cycle of planning, execution, and correction common in decision-making. This structure is particularly valuable in interactive environments. For example, an agent built on DLLMs could generate a tentative plan, refine it in response to feedback, and iteratively converge toward a reliable action sequence. Exploring this agentic potential may bridge the gap between static text generation and dynamic reasoning required in real-world tasks.

## S9 ASSET LICENSE AND CONSENT

All models and datasets used in this work are publicly available. We strictly comply with their original licenses and use them only for non-commercial academic research. The contents of datasets do not represent our views or opinions.

**Models.** We evaluate five open-source models: Llama-3.1-8B (Meta custom license, attribution required, outputs may not be used to train competing models), Mistral-7B (Apache 2.0, permissive), DeepSeek-7B (DeepSeek custom license, attribution required), Dream-7B (Apache 2.0), and LLaDA-8B (MIT license). All licenses permit academic research use; detailed terms are available via the original model repositories.

**Datasets.** We use standard reasoning and QA benchmarks: GSM8K (MIT), MathQA (Apache 2.0), Minerva Math (MIT), ANLI (CC-BY-NC 4.0), MNLI (OANC + CC-BY-SA), RTE (GLUE permissive), LSAT-LR (MIT (via AGIEval)), LogiQA-en (CC-BY-NC-SA 4.0), COPA (CC-BY 4.0), PIQA (MIT), OpenBookQA (CC-BY-SA 3.0 for data, Apache 2.0 for code), and HellaSwag (CC-BY-NC 4.0). We note that some datasets include **non-commercial (NC)** and/or **share-alike (SA)** clauses; our use is strictly for academic purposes in compliance with these restrictions.

**Consent.** Our study does not involve crowdsourcing or human subjects. All results are derived from publicly available models and datasets.

## S10    SOCIAL IMPACT AND LIMITATIONS

Our study contributes to understanding how diffusion-based LLMs differ from autoregressive LLMs in reasoning, highlighting their relative strengths across task types. This provides insights into designing future reasoning models that better align with human-like problem solving.

However, several limitations remain. First, our evaluation focuses on a subset of reasoning tasks (mathematics, logic, commonsense, NLI), while broader domains such as multimodal reasoning (Driess et al., 2023; Achiam et al., 2023) and interactive agents (Wang et al., 2023a; Zeng et al., 2023) are not yet covered. Second, we mainly study mid-sized models (7B–8B), leaving open whether the relative advantages of DLLMs persist or amplify at larger scales (70B–100B). Third, our work does not introduce a concrete hybrid model that integrates AR and diffusion. A promising direction for future research is to design such a pipeline, where an AR LLM seeds a draft that is subsequently refined by diffusion.

Future work should therefore extend task coverage, validate scaling behavior, and develop practical hybrid pipelines that combine the strengths of both paradigms.

## S11    AI DISCLOSURE

We acknowledge the use of GPT-5 for grammar checking only. The model was employed to correct grammatical errors while ensuring the original meaning and intent of the text remained unchanged.

