# OpenReview forum: "Beyond Next-Token Prediction: Diffusion vs. Autoregressive Reasoning in LLMs"
_ICLR.cc/2026/Conference — ICLR 2026 Conference Withdrawn Submission_

### Official Review · Reviewer_au6D · 2025-10-27

**Soundness:** 1
**Presentation:** 2
**Contribution:** 2
**Rating:** 2
**Confidence:** 3

**Summary:**

The work presents a comparative study between autoregressive LLMs (AR LLMs) and diffusion LLMs (DLLMs). With around twenty benchmarks and five LLMs (3 AR LLM and 2 DLLM), the authors draw insights into the advantages of both sides. The paper argues that DLLMs yield higher accuracy on tasks requiring global constraints, while LLMs remain competitive on simpler commonsense QA. Also, DLLMs are more robust to prompt noise and distribution shift. However, DLLMs are generally slower than AR LLMs.

**Strengths:**

- A clear writing flow guided by research questions.
- Extensive benchmarks (~20) provide comprehensive insights to those audiences not familiar with DLLMs.

**Weaknesses:**

- The comparison between DLLM and reasoning AR LLMs is missing.

I see the statement in lines 163-176: "we exclude the results that AR LLMs trained with reinforcement learning (RL) for fair comparisons, as diffusion-based LLMs remain in an early stage and RL approaches for them are not yet sufficiently developed." I disagree with this exclusion.   Early mistakes leading to incorrect reasoning trajectories were a big problem before the RL models, but as highlighted by Deepseek R1, current LLMs with massive RL possess the ability to correct early mistakes in later reasoning traces. RL models represent a huge advancement in AR LLMs and should not be ignored.

- Comparisons are done on base models instead of instruct-tuning models.

All the AR baselines are base models, which makes their results look different from expectations. I checked LLaDA's project page (https://ml-gsai.github.io/LLaDA-demo/) and found that instruction tuning is not as effective for DLLM compared to AR LLM. When comparing instruct models of AR LLM and DLLM, we can see that LLaMA3 is at least the same, if not better, than DLLM, on reasoning tasks. This directly opposes the conclusions drawn by the paper.

- Work is largely empirical, and the writing has slight over-claims.

The entire paper is an empirical study. While this is not by itself a ground for rejection, I imagine researchers in the respective areas would not find significant value in this work if they had experimented with DLLMs before. The author claims an apple-to-apple comparison. This is an overclaim given that the models are trained with different datasets. A true apple-to-apple comparison would require these models to be trained from scratch.

**Questions:**

See the weakness section. In particular, please address the following questions:
1. Do the conclusions and observations presented in this paper still hold for RL models?
2. Given that users always use instruct models, do the conclusions and observations hold for instruct-tuning models?
3. What is the CoT setting in the paper? Why does CoT give worse results for many cases (especially DLLMs)? Does this indicate some problems in DLLMs?
4. (To strengthen the paper,) are there any additional insights (for example, optimization directions for DLLMs with concrete plans, new methods to combine the advantages of two directions, etc.) that can be presented in this work?

---

### Official Review · Reviewer_wtx7 · 2025-10-27

**Soundness:** 2
**Presentation:** 3
**Contribution:** 2
**Rating:** 4
**Confidence:** 4

**Summary:**

This paper presents an extensive empirical comparison between autoregressive (AR) and diffusion-based language models (DLLMs). Through experiments spanning multiple domains, the authors find that DLLMs outperform AR models on the reasoning tasks evaluated, particularly those involving multiple constraints. The paper also includes an investigation into how key hyperparameters influence the reasoning performance of DLLMs.

**Strengths:**

1. The paper addresses a timely and relevant question regarding the comparative performance of AR LLMs and DLLMs. The experimental evaluation is comprehensive, covering multiple model classes and benchmark datasets.

2. The hyperparameter analysis for DLLMs offers practical insights that could be valuable for practitioners working with these models.

3. The paper is generally clearly written and well-organized.

**Weaknesses:**

1. The central concern with this paper is the fairness of the primary experimental comparison. The authors' main findings are based on comparing 8B models from both AR LLM and DLLM families. However, as the paper notes (line 248), DLLMs require approximately 1000x more FLOPs per token, and they also produce slightly longer outputs. This vast disparity in computational budget between the two model types raises significant concerns about the fairness of the comparison and, consequently, the validity of the conclusions drawn from it.

2. While the authors include a comparison under a matched computational budget (Section 5, Finding III), this analysis appears limited. It is conducted on a small subset (100 examples) of GSM8K and restricts the AR methods to multi-time execution and beam search. This limited experiment does not seem sufficient to substantiate the paper's broad claims, which are largely derived from the compute-imbalanced main comparison. The paper's conclusions would be much more convincing if this matched-compute analysis were more thorough.

3. The abstract states that "mechanistic analyses show DLLMs can gradually correct early errors and enforce consistency." However, this mechanistic analysis appears to be missing from the main body of the paper. Expanding on this point would provide valuable insight into why DLLMs exhibit the reported performance, moving beyond what they achieve.

**Questions:**

1. Could the authors provide a clearer justification for the decision to compare models based on parameter count (8B) rather than on a matched computational budget, especially given the ~1000x FLOPs difference?

2. Could the authors elaborate on the mechanistic analysis of error correction in DLLMs mentioned in the abstract, or clarify where this analysis can be found in the text?

---

### Official Review · Reviewer_gFpX · 2025-10-31

**Soundness:** 2
**Presentation:** 1
**Contribution:** 1
**Rating:** 0
**Confidence:** 4

**Summary:**

This paper studies LLM reasoning in autoregressive LLM and diffusion-based LLM. The paper does a comparison between the two types of LLMs and shows that DLLMs outperform AR LLMs on most reasoning benchmarks. The paper also reports that current DLLMs are far less efficient and that CoT prompting helps AR baselines substantially but provides little to no gains for DLLMs.

**Strengths:**

- The paper tries to question the dominance of autoregressive decoding by asking if diffusion models reason better.
- The paper is rich with various experiments and ablations that compare DLLMs and AR LLMs.
- The paper considers both the trade-off between accuracy and the efficiency (measured wwith Flops per token)

**Weaknesses:**

- Lack of novelty: Despite ambitious framing, the paper presents no substantive methodological or conceptual innovation, functioning mainly as a broad empirical comparison rather than an original research advance.
- Scale limitation: All models are mid-sized (7B–8B). Without results at 30B+ or 70B+ scale, it’s unclear whether observed trends persist for competitive LLMs. This hinders generalization.
- Overstated framing: The title and narrative imply a paradigm shift “beyond next-token prediction,” yet the work is essentially a collection of empirical observations comparing existing architectures, without theoretical grounding or deeper analytical insight to justify such framing.
- Unclear practical guidance: Although the paper claims to offer actionable advice on when to use DLLMs versus AR LLMs, the guidance is implicit and limited. Beyond a brief mention in the conclusion, the suggestion to use hybrid AR+Diffusion systems is largely speculative, with little empirical evidence or analysis to substantiate its effectiveness. While the appendix includes a “Future Directions” section, the main paper should be self-contained.

**Questions:**

Check weaknesses above.

---

### Official Review · Reviewer_LaRC · 2025-11-01

**Soundness:** 3
**Presentation:** 3
**Contribution:** 3
**Rating:** 6
**Confidence:** 3

**Summary:**

This paper compares autoregressive LMs and discrete diffusion LMs (DLLMs) across several axes and reports useful findings about when to use each and how to set DLLM hyperparameters. Broadly the two strategies are complementary: DLLMs tend to do better on tasks requiring dense mutual constraint satisfaction, while AR LMs are mildly better on commonsense QA and much better on latency. On GSM8K, DLLMs retain an advantage under compute matching on a small subset, and they are more robust to prompt noise.

**Strengths:**

Clear thesis with practical value: DLLMs win on tasks with deeply interdependent constraints, at the cost of latency. Experiments are reasonable, ablations are informative, Table 4’s compute matching addresses a central concern, and the writing is clear (if occasionally repetitive).

**Weaknesses:**

The central claim would benefit from a principled interdependence metric and a synthetic benchmark (e.g., graph coloring with controllable degree or treewidth) to exhibit a crossover where DLLMs surpass AR as coupling increases. A similar analysis could separate genuinely “global coupling” from forced-move phases (e.g., late Sudoku, unit propagation in SAT).

**Questions:**

Can the notion of interdependence be made precise—for example, with an information-theoretic measure of average coupling between token sets? This may matter for datasets that are neither clearly mathematical nor purely commonsense.

---

### Note · Authors · 2026-01-05

I have read and agree with the venue's withdrawal policy on behalf of myself and my co-authors.